# Non-Contact Measurement of Motion Sickness Using Pupillary Rhythms from an Infrared Camera

**DOI:** 10.3390/s21144642

**Published:** 2021-07-06

**Authors:** Sangin Park, Sungchul Mun, Jihyeon Ha, Laehyun Kim

**Affiliations:** 1Center for Bionics, Korea Institute of Science and Technology, Seoul 02792, Korea; sipark@kist.re.kr (S.P.); hj910410@kist.re.kr (J.H.); 2Department of Industrial Engineering, Jeonju University, Jeonju 55069, Korea; sungchul.mun@jj.ac.kr; 3Department of Biomedical Engineering, Hanyang University, Seoul 04673, Korea

**Keywords:** motion sickness, pupillary rhythms, cognitive load, non-contact measurement

## Abstract

Both physiological and neurological mechanisms are reflected in pupillary rhythms via neural pathways between the brain and pupil nerves. This study aims to interpret the phenomenon of motion sickness such as fatigue, anxiety, nausea and disorientation using these mechanisms and to develop an advanced non-contact measurement method from an infrared webcam. Twenty-four volunteers (12 females) experienced virtual reality content through both two-dimensional and head-mounted device interpretations. An irregular pattern of the pupillary rhythms, demonstrated by an increasing mean and standard deviation of pupil diameter and decreasing pupillary rhythm coherence ratio, was revealed after the participants experienced motion sickness. The motion sickness was induced while watching the head-mounted device as compared to the two-dimensional virtual reality, with the motion sickness strongly related to the visual information processing load. In addition, the proposed method was verified using a new experimental dataset for 23 participants (11 females), with a classification performance of 89.6% (*n* = 48) and 80.4% (*n* = 46) for training and test sets using a support vector machine with a radial basis function kernel, respectively. The proposed method was proven to be capable of quantitatively measuring and monitoring motion sickness in real-time in a simple, economical and contactless manner using an infrared camera.

## 1. Introduction

The development and generalization of head-mounted devices (HMDs) has made virtual reality (VR) a real-life experience. VR technology has been extended to various applications, such as architecture, education, training, mobile devices, medical visualization, interactions, entertainment and manufacturing [1,2,3]. Positive effects have been reported regarding the increase in efficiency of work tasks and the ability to experience a real presence and coexistence [4,5,6,7]. However, the side effects of motion sickness have been widely reported by some users who have used flight and driving simulators and many other virtual environments [8,9,10,11,12], with common symptoms including visual fatigue, anxiety, nausea and disorientation apart from abdominal and oculomotor symptoms [13,14,15,16]. Visually induced motion sickness is caused by incongruities in the spatiotemporal relationships between actions (such as hand movements) and perceptions such as corresponding visual feedback, which leads to distortions and delays in the visual information system [17]. As these issues are a major obstacle for the further development of the VR industry, research to understand and resolve these issues is required to improve the VR experience for viewers [18,19,20].

The symptoms of motion sickness in VR are known to be caused by a variety of factors, such as gaze angle, fixation, retinal slip and the field of view of the HMD [21,22,23,24]. The relationship between these causal factors and motion sickness need to be verified, and there is a need to provide guidelines for content/device developers and users to minimize the symptoms so that users may be comfortable and enjoy the VR contents. Many previous studies have tried to measure motion sickness using various human responses with the following measurement tools: (1) subjective ratings such as a simulator sickness questionnaire (SSQ) [13,25,26,27], a motion sickness susceptibility questionnaire (MSSQ) [21,28,29,30,31,32], a Coriolis test [33,34] and a questionnaire developed by Graybiel and Hamilton [35]; (2) behavioral responses such as head motions [25], body movement [21,36] and eye blinking [28]; (3) autonomic nervous system (ANS) responses such as heart rate (HR) [28,30,33,34,37], autonomic balance [21,26,32,33,34,35], skin temperature (SKT) [28], galvanic skin response (GSR) [28], respiration (RR) [26,28] and blood pressure (BP) [26]; (4) central nervous system (CNS) responses such as an electroencephalogram (EEG) spectrum [31] and functional magnetic resonance imaging (fMRI) [32].

However, each of these measurements had limitations. The subjective measurement of experiences using questionnaires can be impacted by individual differences, depending on personal interpretations and experiences [38,39] meaning that other measures were required to solve these individual differences. Measurement using behavioral responses is tasked with determining a physiological mechanism for motion sickness that does not trigger a more general response. Physiological and neurological responses such as electrocardiogram (ECG), photoplethysmography (PPG), GSR, SKT and EEG have significant disadvantages both in the measurement burden of sensor contact with the skin and the need for an additional device for data acquisition. In our study, the non-contact measurement of motion sickness was developed by processing data from the pupillary responses obtained using an infrared (IR) camera. The camera-based pupillary measurement has practical applicability in HMDs to measure motion sickness, without requiring other additional devices. Our previous study confirmed that the motion sickness from HMDs causes significant changes in pupil rhythms. After experiencing the motion sickness, the pupillary rhythms revealed the irregular patterns with the increasing values in the mean and standard deviation of the pupil diameter. This phenomenon can be interpreted as the cognitive load caused by the increasing volume of visual information and the sensory conflict [40]. The purpose of this work is to develop a real-time system that can monitor the motion sickness based on new features.

The cause of the motion sickness can be interpreted by the “sensory conflict theory”. Following this theory, motion sickness is caused by a conflict, or inconsistency, between different sensory modalities, such as vestibular and visual information [41]. For example, the users experienced motion sickness when there was conflict or inconsistency between the visual information of VR content and the corresponding bodily feedback. This motion sickness has been strongly correlated with the decay of information processing in the brain, such as cognitive load or mental workload. Previous studies have reported that three-dimensional 3D visual fatigue is related to cognitive load rather than visual discomfort or eye strain. As 3D VR imaging involves more visual information, such as depth, than 2D images, it requires greater brain capacity, or resources, to process visual information [19,20,42,43]. In the case of motion sickness, the experience of VR content using an HMD may be required for neural resources because the VR content involves more visual information than 2D. In addition, owing to the motion sickness being related to the inconsistency between visual and vestibular information, this phenomenon can accelerate the load of visual information processing. Thus, the motion sickness is attributed to an increase in the amount of visual information to be processed and to the loss of neural resources caused by the inconsistency among different sensory information, which is interpreted as a high-level cognitive load.

Both physiological (i.e., the sympathetic and parasympathetic nervous systems) and neurological (i.e., brain functions such as memory, attention, cognition, perception and affective processing) are reflected in the pupillary rhythm via neural pathways (both afferent and efferent pathways) between the brain and the pupil nerves [44,45,46,47,48]. In particular, the pupillary rhythm has been observed to be involved in cognitive function among neurological mechanisms such as cognitive load or mental workload [49,50,51,52], attention [53,54] and working memory [49,50]. Thus, this study aimed to interpret the phenomenon of motion sickness using the cognitive load mechanisms reflected by the pupillary rhythm and to develop an advanced non-contact measurement method, via an infrared camera, for measuring motion sickness.

## 2. Materials and Methods

### 2.1. Experimental Design

Thirty-two volunteers applied for this experiment, participating in the pretask to measure their sensitivity to motion sickness. The participants were completely focused on the VR contents of “Ultimate Booster Experience” (GexagonVR, 2016) through HTC VIVE (HTC Inc., Taoyuan City, Taiwan) for 10 min, and then asked to report their motion sickness. Eight volunteers who did not experience motion sickness were excluded from the main experiment. Twenty-four healthy subjects participated in the experiments (12 females, all right-handed, average age of 24.34 ± 2.06 years). We recruited the volunteers with the specific conditions as follows: (1) normal vision or corrected-to-normal acuity (i.e., over 0.8) and (2) no medical and family history associated with their visual function or autonomic or central nervous system. They were required to get enough sleep the day before the experiment and to abstain from alcohol, cigarettes and caffeine for 12 h to minimize the negative effects of fatigue, autonomic and central nervous function. The study was approved by the Institutional Review Board of Sangmyung University, Seoul (BE2017-21). All participants signed informed consent forms before their participation.

This study was designed by a “within-subjects design” to compare the viewer’s experience for the VR contents from 2D (non-motion sickness) and HMD (motion sickness) devices. Participants experienced the VR content using either the 2D or the HMD version of the VR content from “NoLimits 2 Roller Coaster Simulation” (Ole Lange, Mad Data GmbH & Co. KG, Erkrath, NRW, Germany, 2014) for 15 min on the first day, and on the next day, they watched the VR content in the other version (i.e., first day HMD and the second day, 2D with the order randomized across subjects). For watching the VR contents for both the versions (2D and HMD), each participant used a 27-inch LED monitor (27MP68HM, LG) and HTC VIVE (HTC Inc., New Taipei City, Taiwan, and Valve Inc., Bellevue, WA, USA), respectively. Before and after viewing the VR content, subjective ratings based on an SSQ were evaluated, and the participants’ pupil images were recorded for 5 min. The subjective ratings and pupillary responses before and after the simulation were compared. The setup of the experimental procedure and environment is shown in Figure 1.

The SSQ was selected for 16-items for the motion sickness from the motion sickness questionnaire (MSQ), and categorized into three non-mutually exclusive factors using the factor analysis based on the relationship between SSQ items, namely: nausea (*N*), oculomotor responses (*O*) and disorientation (*D*). All three factors were each further divided into seven general items. Nausea consisted of general discomfort, increased salivation, sweating, nausea, difficulty concentrating, stomach awareness and burping. Oculomotor responses items consisted of general discomfort, fatigue, headache, eyestrain, difficulty focusing, difficulty concentrating and blurred vision, and disorientation consisted of difficulty focusing, nausea, fullness of head, blurred vision, dizziness (eyes open), dizziness (eyes closed) and vertigo [17]. The examples of SSQ are shown in Appendix B. Participants reported their experience with motion sickness using a 4-point scale (0–3) for 16 questionnaires, and the total SSQ score was calculated using Equation (1) [17], where the values of *N*, *O* and *D* were defined by summing the rating values of each questionnaire for nausea, oculomotor responses and disorientation, respectively. Example of SSQ are shown in Appendix B.
(1)Total SSQ score={(N×9.54)+(O×7.58)+(D×13.92)}×3.74

### 2.2. Data Acquisition and Signal Processing

The pupil images were recorded at 30 fps with a resolution of 960 × 400 (pixels) using a GS3-U3-23S6M-C IR camera from Point Grey Research Inc., Richmond, BC, Canada. In addition, an infrared lamp (Genie Compact IR LED Illuminator, 30-degree, 850 nM wavelength and 20 m IR range) was used to detect the pupil area. Since changes in ambient light can affect the pupillary response, the ambient light of experimental room was controlled from 150 to 170 lx (163.42 ± 7.14 lx) measured by the Visible Light SD Card Logger (Sper Scientific Meters Ltd., Scottsdale, AZ, USA) at a 2 Hz sampling rate. The signal processing to extract the pupillary response was conducted based on methods from previous studies [55,56,57,58] as follows. First, the input eye images (gray scale) from the IR camera were processed by binarization based on a threshold value that was reported in previous studies established using a linear regression model between the mean and maximum brightness values from the entire image [56,57,58], as shown in Equation (2).
(2)$$Threshold\;value=\left(-0.418\times B_{mean}\right)+\left(1.051\times B_{max}\right)+7.973$$
where *B_mean_* and *B_max_* denote the brightness value of the mean and maximum of the entire image on a gray scale, respectively. Second, the pupil area was detected using a circular edge detection (CDE) algorithm [55], as shown in Equation (3). If the multiple pupil position was selected, the position closest to the reflected light caused by the infrared lamp was selected to accurately detect the pupil position.
(3)$${\mathrm{Max}}_{\left(r,{\;x}_{0},{\;y}_{0}\right)}\left|G\sigma \left(r\right)*\frac{\partial }{\partial r}{\oint^{\;}}_{r,{\;x}_{0},{\;y}_{0}}\frac{I\left(x,y\right)}{2\pi r}\mathrm{ds}\right|$$
where I(x,y) indicates the gray level at the (x, y) position, ($x_{0},{\;y}_{0}$) and r represents the center position and radius of the pupil, respectively. Finally, the pupil diameter (pixel) was extracted by detecting the pupil area, as shown in Figure 2.

The procedure of signal processing and indicator definitions for the pupillary rhythm for detecting motion sickness are shown in Figure 2. (1) After detecting the pupil area, the pupil diameter was calculated using the number of pixels, see Figure 2A,B. (2) The pupil diameter was used to process the sliding movement average (i.e., a 1 s window size and a 1 s resolution) from 30 to 1 fps to minimize the effect of eye closure, and this signal defined the pupillary rhythm in this study, see Figure 2C. The pupil diameter was not calculated if the pupil area was not detected when the eyes were closed. This method was used to acquire the pupil diameter, based on this procedure for the sliding movement average, if a participant took less than a second to blink. (3) To determine whether the patterns of pupil rhythm were regular or irregular, the mean (mean of PD signals, mPD) and standard deviation (SD of PD signals, sPD) were calculated from the pupillary rhythm and were defined as indicators of motion sickness, see Figure 2D. (4) The pupillary rhythm was then processed by a fast Fourier transform (FFT) based on the Hanning window technique to extract the spectral information, see Figure 2E. (5) The ratio of the power of dominant peaks in the entire band was calculated and defined as the pupillary rhythm coherence (PRC) ratio based on metrics used in previous research [59], as shown in Equation (4). Increasing the PRC value was interpreted by the fact that the pupillary rhythm generally is stable at a certain frequency (dominant peak frequency) band, and vice versa. The dominant peak was identified in the range of 0–0.5 Hz, and its power was extracted. The total power was the sum of all power values in the range of 0–0.5 Hz, see Figure 2F.
(4)$$\mathrm{PRC}\;\mathrm{ratio}=\frac{\;\mathrm{Power}\;\mathrm{of}\;\mathrm{Domonant}\;\mathrm{Peak}\;\mathrm{Band}}{\left(\mathrm{Power}\;\mathrm{of}\;\mathrm{Total}\;\mathrm{Band}\;-\mathrm{Power}\;\mathrm{of}\;\mathrm{Domonant}\;\mathrm{Peak}\;\mathrm{Band}\right)}$$

### 2.3. Statistical Analysis

This study was designed using a “within subject design” and the motion sickness responses of individual subjects to 2D and HMD content were compared. Thus, in the statistical analysis, a paired *t*-test was selected based on the normality test to compare the pupillary response before and after viewing each condition. In addition, an analysis of covariance (ANCOVA) was also applied to compare the pupillary responses to the 2D and HMD conditions, because the independent *t*-test could not confirm the viewer’s state before watching the VR content. The ANCOVA compared dependent variables (post-viewing content) between groups, with the pre-viewing content baseline as a covariate [19,43,60]. A partial correlation was used to analyze the correlation between SSQ scores and pupillary responses (post-viewing contents) in all conditions (2D and HMD), considering the pre-viewing contents as covariates [61]. A Bonferroni correction was then applied to resolve the problem of type I errors caused by multiple comparisons, and the statistical significance was controlled based on the numbers from each individual hypothesis (i.e., α = 0.05/*n*) [62,63]. The statistical significance level of indicators for pupillary response was set to 0.0167 (mPD, sPD and PRC ratio, α = 0.05/3). In addition, this study applied the effect size to verify the practical significance based on Cohen’s d with a *t*-test and the partial eta-squared value (ƞp^2^) with an F-test. The standard values for practical significance of 0.10/0.01, 0.25/0.06 and 0.40/0.14 (Cohen’s d/partial eta-squared) were generally regarded as small, medium and large, respectively [64]. All statistical analyses (i.e., paired-samples *t*-test, ANCOVA and partial correlation) were conducted using IBM SPSS Statistics 21.0, for Windows (SPSS Inc., Chicago, IL, USA).

### 2.4. Classification

Four basic machine learning algorithms were used to classify motion sickness (HMD) and normal state (2D)—linear discriminant analysis (LDA) (data standardization), decision tree (DT) (split criteria: maximum deviance reduction; number of splits: 4), linear kernel support vector machine (linear SVM) (data standardization, box constraint: 7.7) and radial basis function kernel support vector machine (RBF-SVM) (data standardization, box constraint: 0.09) [65,66,67,68]. Three pupillary features, namely the mPD, the sPD and the PRC ratio, were extracted from the experimental data, and the three statistical features showing statistically significant results were trained by the four classification algorithms on a 24-subject dataset with ten-fold cross validation. The classification performances of the four algorithms were evaluated based on their area under the curve (AUC) for receiver operating characteristics (ROC), accuracy, sensitivity, and specificity [69,70]. A new dataset of 23 subjects (11 females), with ages ranging from 23 to 29 y (mean age 25.02 ± 3.14), was also applied to trained classification models to evaluate practical performance. The statistics and machine learning toolbox of Matlab (2019b, Mathworks Inc., Natick, MA, USA) was used for classification and cross-validation. The classification measures are defined as follows.

Accuracy is used to calculate the proportion of the total number of predictions that are correct.

Accuracy (%) = (TP + TN)/(TP + FN + TN + FP) × 100

Sensitivity is used to measure the proportion of actual positives that are correctly identified.

Sensitivity (%) = TP/(TP + FN) × 100

Specificity is used to measure the proportion of actual negatives that are correctly identified.

Specificity (%) = TN/(TN + FP) × 100

AUC: area under the receiver operating characteristic curve. The AUC value lies between 0.5 and 1, where 0.5 denotes a bad classifier and 1 denotes an excellent classifier.

Here, TP represents the correctly classified motion sickness (HMD), FN is the incorrectly classified motion sickness (HMD), TN is the number of true negative classifications and FP is the number of true positive classifications.

## 3. Result

### 3.1. SSQ Scores

The paired-samples *t*-test showed significant differences (increasing the post-viewing) for the total SSQ score in the HMD viewing conditions between the pre- and post-viewing (t(46) = −14.640, *p* = 0.0000, with a large effect size (Cohen’s d = 4.317)). In the 2D viewing condition, no significant differences were found between the pre- and post-viewing conditions (t(46) = −0.805, *p* = 0.4041). The ANCOVA analysis showed significant differences (increasing the HMD condition) in the post-viewing condition for the total SSQ score with the pre-viewing condition as a covariate (F(1,46) = 149.035, *p* = 0.0000, with a large effect size (ƞp^2^ = 0.768)), as shown in Figure 3.

### 3.2. Pupillary Response: Time Domain Index

As shown in Figure 4, the pupillary rhythms were fairly regular and stable in the pre- and post-viewing 2D conditions, with the pupil diameters almost identical. After being exposed to the 2D condition, the mean pupil diameters (mPDs) for participants 1, 10 and 24 were changed from 35.556, 34.853 and 37.219 to 34.004, 34.872 and 38.467 pixels, with the changes in standard deviations (sPDs) from 1.276, 0.968 and 1.549 to 1.264, 1.001 and 1.332 pixels, respectively. In contrast, participants’ pupillary rhythms were fairly regular and stable before the HMD viewing condition. However, they became irregular and unstable after the HMD viewing condition with the significantly increasing pupil diameter. After the HMD viewing condition, the mPD values for participants 1, 10 and 24 were changed from 35.168, 35.465 and 36.342 to 42.522, 44.885 and 43.620 pixels, with the following changes in their sPD values from 1.276, 1.046 and 1.564 to 2.889, 2.649 and 2.417 pixels. Similar results were observed for most of the participants, except for participant 6 showing no significant differences in the mPD and sPD values between the 2D and the HMD viewing conditions. The mPD and sPD values for participant 6 were changed from 39.902 and 0.854 to 38.200 and 0.798 pixels after experiencing the 2D condition, and changed from 39.543 and 0.798 to 38.899 and 0.902 pixels after experiencing the HMD condition.

A paired-sample *t*-test showed significant differences (increasing during the post-viewing measurement) for the mPD and sPD in the HMD viewing condition between the pre- and post-viewing measurements (mPD: t(46)) = −11.544, *p* = 0.0000, with a large effect size (Cohen’s d = 3.404); sPD: t(46)) = −8.265, *p* = 0.0000, with a large effect size (Cohen’s d = 2.437). In the 2D viewing condition, no significant differences were found between the pre- and post-viewing measurements (mPD: t(46) = −0.645, *p* = 0.5251; sPD: t(46) = −2.156, *p* = 0.0418]) based on the Bonferroni correction. The ANCOVA analysis showed a significant difference (increasing after the HMD condition) in the post-viewing condition for mPD and sPD with the pre-viewing condition as a covariate (mPD: F(1,46) = 90.793, *p* = 0.0000, with a large effect size (ƞp^2^ = 0.669); sPD: F(1,46) = 37.248, *p* = 0.0000, with a large effect size (ƞp^2^ = 0.453)), as shown in Figure 5 and Appendix A.

### 3.3. Pupillary Response: Frequency Domain Index

As seen in Figure 6, the spectral power of the pupillary rhythms was concentrated in a specific frequency band before and after viewing, for the 2D condition. For example, the PRC ratios for participants 1, 10 and 24 changed from 0.530, 0.467 and 0.579 to 0.556, 0.429 and 0.500, respectively, after experiencing the 2D condition. In contrast, the spectral power of pupillary rhythms was concentrated in a specific frequency band in the pre-viewing measurement for the HMD condition, but dispersed across the entire frequency band after viewing. The PRC ratios for participants 1, 10 and 24 changed from 0.595, 0.473 and 0.605 to 0.092, 0.043 and 0.072, respectively, after experiencing the HMD condition. These results were reported for most of the participants, however, participant 6 again showed no significant difference between the 2D and HMD viewing. The PRC ratio for participant 6 changed from 0.505 to 0.481 after experiencing the 2D condition and changed from 0.505 to 0.476 after experiencing the HMD condition.

A paired-samples *t*-test showed significant differences (decreasing the post-viewing) in the PRC ratio in the HMD viewing condition between the pre- and post-viewing measurements (t(46) = 10.483, *p* = 0.0000, with a large effect size (Cohen’s d = 3.091)). In the 2D viewing condition, no significant differences were found between the pre- and post-viewing measurements (t(46) = 2.341, *p* = 0.0283) based on the Bonferroni correction. The ANCOVA analysis showed a significant difference (decreasing after the HMD condition) in the post-viewing condition for the PRC ratio with the pre-viewing condition as a covariate (F(1,46) = 75.358, *p* = 0.0000, with a large effect size (ƞp^2^ = 0.629)), as shown in Figure 7 (see Table A1).

### 3.4. Correlation Analysis and Classification

A multiple regression analysis was conducted for partial correlation and for calculating the residuals with covariates (SSQ scores and pupillary responses in the pre-viewing conditions). As seen in Figure 8, the plot for residuals of SSQ scores and pupillary responses (mPD, sPD and PRC ratio) with linear regression lines. The correlation coefficients between SSQ scores and each pupillary response in the post-viewing condition were statistically significant (mPD: r = 0.751, *p* = 0.0000; sPD: r = 0.559, *p* = 0.0000; PRC ratio: r = −0.756, *p* = 0.0000).

As seen in Table 1, classification measures existed (accuracy, sensitivity, specificity and AUC) for the training dataset with 10-fold cross validation and for the new dataset according to four classifiers (LDA/decision tree/linear SVM/RBF SVM). ROC curves were applied for the training dataset in Figure 9A and for the test dataset in Figure 9B and Appendix A.

### 3.5. Non-Contact Measurement System of Motion Sickness in Real Time

The real-time system for the noncontact measurement of motion sickness in this study consisted of an HMD device, add-on IR camera (HTC Vive Eye Tracking Add-On from Pupil Labs, 120 fps with 640 × 480 resolution), add-on lamp and a personal computer for analysis, and can be classified as motion sickness or non-motion sickness state using non-contact measurement, as shown in Figure 10. This system was developed using Visual C++ 2010 and OpenCV 2.4.3, and signal processing was performed using LabVIEW 2010 (National Instruments Inc., Austin, TX, USA). The flowchart and non-contact real-time system of motion sickness are shown in Figure 10 and Figure 11, respectively.

## 4. Discussion

The aim of this study was to determine a method for measuring the motion sickness that appears as a side effect of experiencing VR content (HMD) using the pupillary rhythm and to propose a new indicator for evaluating motion sickness (high-level cognitive load). VR content from an HMD was presented to participants with the goal of causing motion sickness, and the pupillary responses of the participants were compared to the responses after a 2D experience. Participants’ responses to an SSQ confirmed their experience of motion sickness from the HMD; such confirmation verified that the changes in their pupillary response were related to motion sickness.

Overall, the study yielded two significant findings: firstly, the pupil diameters significantly increased during motion sickness. Many previous studies have reported that an increased pupil diameter is closely related to a decay in information processing by the brain [49,50,51,71,72]. The increase in pupil diameter in this study provides evidence that the experience of motion sickness is associated with physiological changes in cognitive load.

Second, the standard deviation of the pupillary rhythm significantly increased, and the PRC ratio significantly decreased after experiencing the HMD condition as compared to the 2D condition. An increase in the sPD and a decrease in the PRC ratio revealed irregular changes in pupil size due to the power of the pupillary rhythm spectrum being dispersed across various spectral bands and the deviation of the pupillary rhythms being increased. These results indicate that fluctuations in pupillary rhythms became irregular after experiencing motion sickness. In previous research, cognitive load has been related to heart rhythm patterns (HRPs) with one study reporting that increasing cognitive load leads to a pattern of irregular and unstable heart rhythms [19]. As the heart responds to external sensory inputs such as visual information transmitted to the brain through afferent pathways, cognitive processes occur not only in the brain, but also through brain–heart connectivity, which influences the cognitive function [59,73]. Pupillary rhythms (i.e., change in pupil size) are strongly affected by the regulation of the sympathetic and parasympathetic nervous system (autonomic balance) based on the contraction function of the sphincter and dilator muscles, and the autonomic balance is determined by the HRP [45,47,48,74]. If a pattern of irregular and unstable heart rhythms is related to cognitive load, the irregular rhythm of the pupil can also be interpreted as being related to cognitive load. Additionally, the pupils are known to be closely related to the central nervous system [44,45,46,47,48], and many studies have reported that they are indicators of cognitive load [49,50,51,71,72].

Changes in pupillary rhythms were correlated with functional brain processing, such as cognitive load or mental workload, attention and working memory based on neural pathways in the midbrain. Many previous studies have demonstrated that changes in pupillary rhythms are correlated with neural activity in the locus coeruleus–norepinephrine (LC–NE) system [75,76,77,78,79], dorsal attention network (DAN) (i.e., activity in the superior colliculus and the right thalamus) [79,80,81] and cingulate cortex [79,82]. These regions are known to be related to cognitive and attentional functions. Thus, the neural resources needed to process the visual information in the brain is reflected in the change of pupillary rhythms, and these results support the findings that increase the pupil diameter and show an irregular pattern of pupillary rhythms.

From these two significant findings, the main contributions of this work can be summarized as follows: firstly, an increase in pupil diameter and an irregular rhythm of the pupil are strongly related to motion sickness, which can be interpreted as a decay in the human vision system. Many previous studies reported that 3D visual fatigue is related to the degradation of the human vision system caused by information processing rather than to visual discomfort, because 3D content involves more visual information, such as image depth, than 2D content [14,19,42,83]. Experience of VR content using HMD should also be interpreted as consuming the neural resources to process the massive visual information. Other research has shown that motion sickness from HMD devices is caused by incongruities in the spatiotemporal relationships between actions and perceptions of visual information, which can lead to distortions and delays in the visual information system [17,83]. Thus, motion sickness is related to an increase in visual information to be processed and to the loss of neural resources caused by the inconsistency or conflict among different sensory information, that is, the high-level cognitive load caused by the massive and inefficient information processing. Results show that evaluating the pupils can be an appropriate way to measure motion sickness rather than interpreting symptoms such as dizziness, fatigue and nausea.

Among the algorithms for classifying motion sickness, the RBF–SVM in this study achieved the highest average recognition accuracy (89.6% for training and 80.4% for the test set). To better illustrate the study findings, this study compared the methods and results with those of the past studies, with the accuracy rate of recognizing motion sickness being 79.6–99.6% in the training set and 72.7% in the test set, as shown in Table 2. The majority of previous studies have reported measurement methods for motion sickness using neurophysiological responses such as the electroencephalogram (EEG), however, these methods have limitations such as complex and expensive equipment, inconveniences and a burden of sensor attachment [56,57,58]. In terms of accuracy, sample size, validation data set and usability, these methods outperformed existing state-of-the-art classification methods for motion sickness detection. In this way, motion sickness can be measured by an infrared webcam through a simple, low-cost and non-contact method based on pupillary rhythms.

## 5. Conclusions

The aim of this study was to develop an accurate non-contact measurement method for detecting motion sickness using pupillary rhythms measured with an infrared webcam. This study found that motion sickness was significantly related to the irregular pattern of pupillary rhythms, as demonstrated by increasing mPDs and sPDs and a decreasing PRC ratio. These phenomena can be interpreted as a decay in visual information processing (i.e., a high-level cognitive load). In addition, when it comes to VR using HMDs, monitoring pupillary responses in real time was proven to be more appropriate than examining other behavioral responses because a user’s face is covered by the device. The proposed method can be adopted to quantitatively measure motion sickness using various parameters such as gaze angle, fixation, retinal slip and field of view, and consequently improve the viewing environment of viewer-friendly VR. The list of abbreviation of the manuscript can be found in abbreviations.

## Figures and Tables

**Figure 1 sensors-21-04642-f001:**
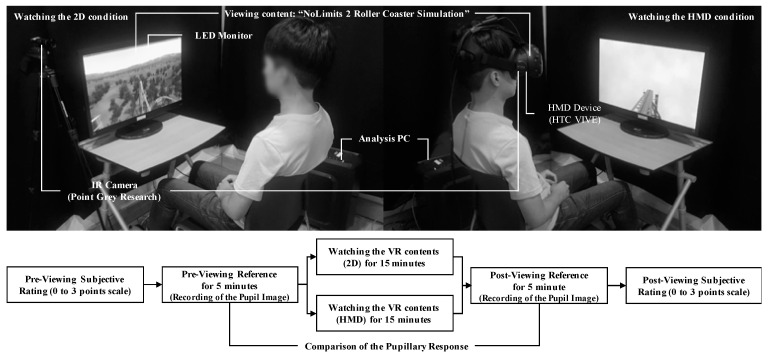
The experimental procedure and environment.

**Figure 2 sensors-21-04642-f002:**
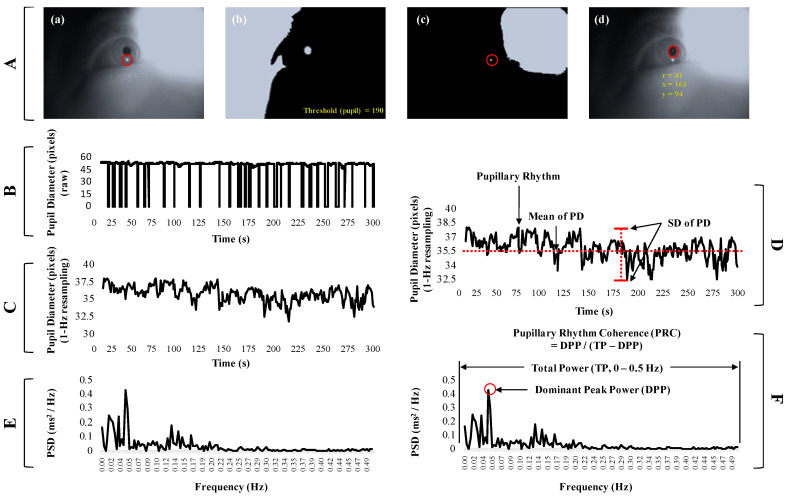
Signal processing for detecting motion sickness from the pupillary rhythm. (**A**) Procedure to detect the pupil area: (**a**) a raw image (gray scale) from the IR camera; (**b**) the binarization image based on the auto threshold; (**c**) detection of the reflected light caused by the infrared lamp; (**d**) detection of the pupil area using the CDE algorithm. (**B**) Pupil diameter signals at 30 fps. (**C**) Resampled pupil diameter (pupillary rhythm) at 1 Hz based on the sliding movement average (window size: 30 fps and resolution: 30 fps). (**D**) Definition for mean and standard deviation (SD) of pupil diameter. (**E**) Spectral signals of pupillary rhythm using the fast Fourier transform (FFT) analysis. (**F**) Definition for pupillary rhythm coherence (PRC).

**Figure 3 sensors-21-04642-f003:**
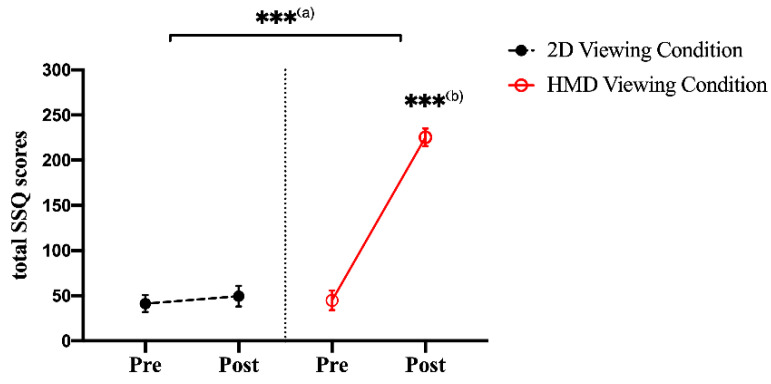
Representation of the total SSQ scores for motion sickness between the 2D and HMD conditions. There was a significant difference based on a paired *t*-test and ANCOVA (*** *p* < 0.001). (**a**) The ANCOVA test between the 2D and HMD viewing condition. (**b**) A paired *t*-test between pre- and post-viewing in the HMD condition.

**Figure 4 sensors-21-04642-f004:**
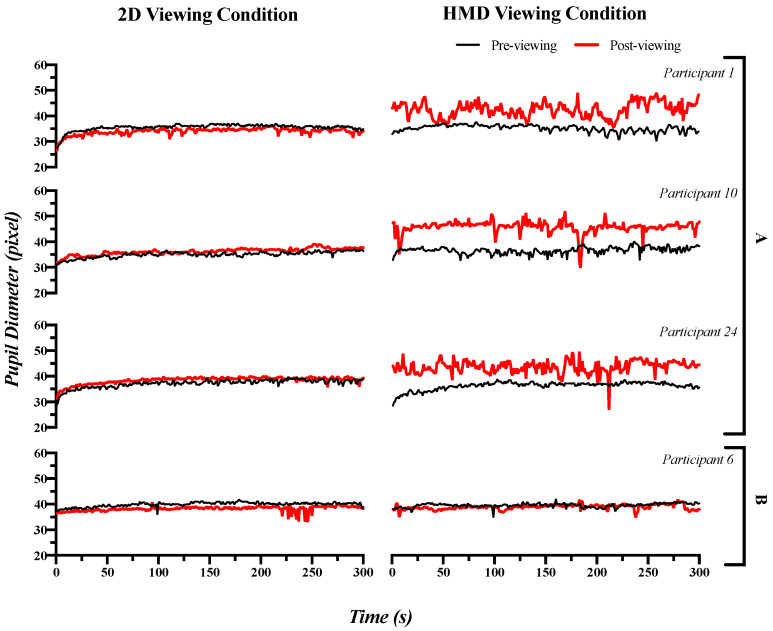
(**A**) Clear examples (for participants 1, 10 and 24) and (**B**) unclear example (for participant 6) of changes in pupillary rhythms (mPD and sPD) pre- and post-viewing 2D and HMD.

**Figure 5 sensors-21-04642-f005:**
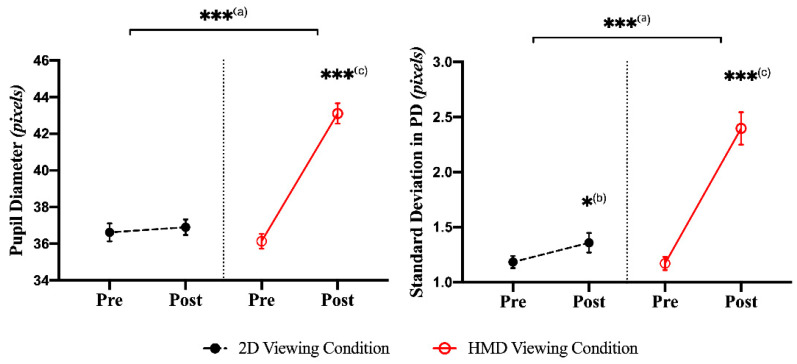
Representation of the mPD and sPD for motion sickness between the 2D and HMD conditions. There was a significant difference based on a paired *t*-test and ANCOVA (* *p* < 0.05; *** *p* < 0.001). (**a**) The ANCOVA test between the 2D and HMD viewing condition. (**b**) A paired *t*-test between pre- and post-viewing in the 2D condition. (**c**) A paired *t*-test between pre- and post-viewing in the HMD condition.

**Figure 6 sensors-21-04642-f006:**
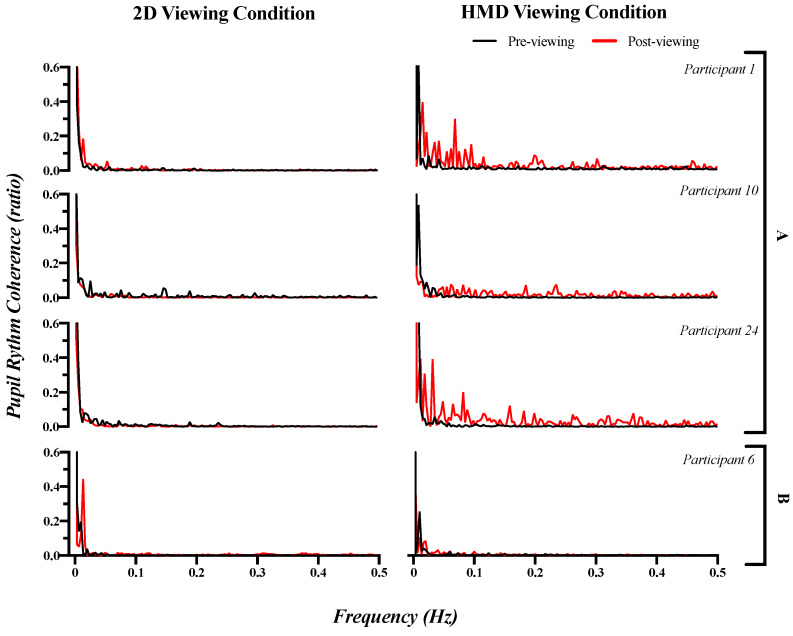
(**A**) Clear (for participants 1, 10 and 24) and (**B**) unclear examples (for participant 6) of changes in the spectrum of pupillary rhythms (PRC ratio) pre- and post-viewing 2D and HMD.

**Figure 7 sensors-21-04642-f007:**
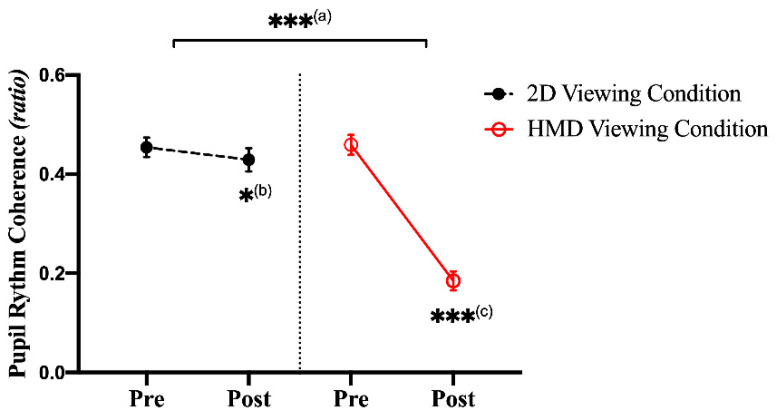
Representation of the PRC ratio for motion sickness between the 2D and HMD conditions. There was a significant difference based on a paired *t*-test and ANCOVA (* *p* < 0.05; *** *p* < 0.001). (**a**) The ANCOVA test between the 2D and HMD viewing condition. (**b**) A paired *t*-test between pre- and post-viewing in the 2D condition. (**c**) A paired *t*-test between pre- and post-viewing in the HMD condition.

**Figure 8 sensors-21-04642-f008:**
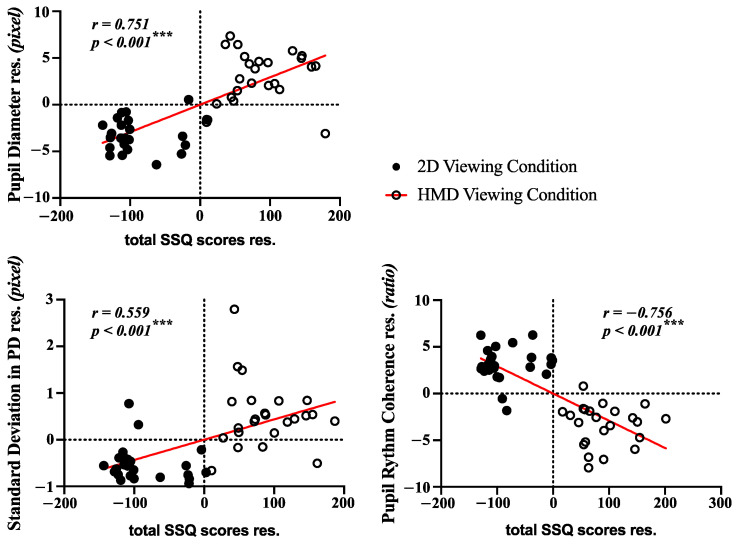
Results of the correlation analysis between SSQ scores and significant features of pupillary rhythms.

**Figure 9 sensors-21-04642-f009:**
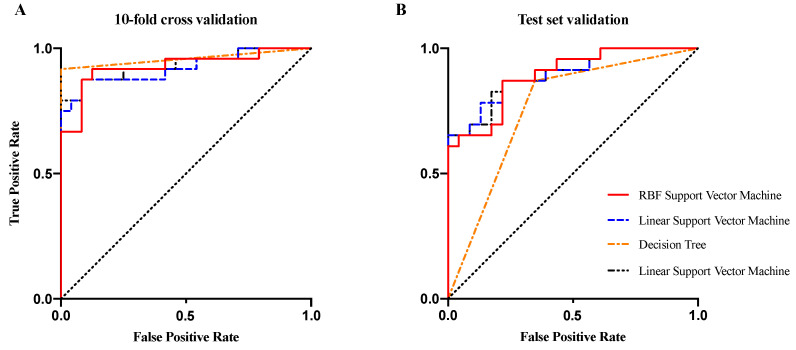
Receiver operating characteristics curves for the (**A**) training dataset and (**B**) test dataset according to the four classifiers.

**Figure 10 sensors-21-04642-f010:**
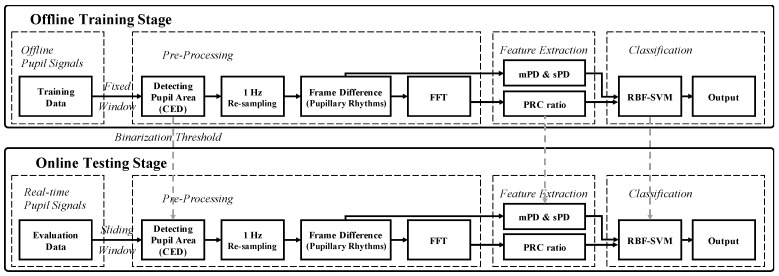
Flowchart for the offline training and online testing stage in the non-contact measurement system of motion sickness.

**Figure 11 sensors-21-04642-f011:**
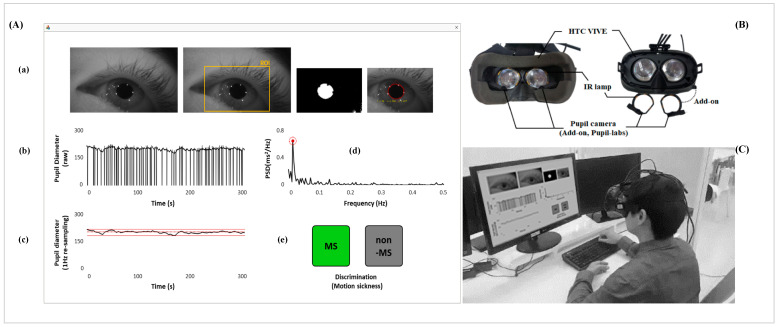
Non-contact measurement system of motion sickness using an infrared (IR) webcam. (**A**) Introduction to the measuring software: (**a**) protocol for detecting the pupil area; (**b**) raw signals of pupillary diameter; (**c**) filtered pupil diameter signals in time domain and detecting _M_PD and _S_PD; (**d**) power spectral density of pupillary rhythms in the frequency domain and detecting the PRC ratio; (**e**) binary decision for the motion sickness state. (**B**) Configuration of the measuring device including the HMD device, add-on IR webcam and lamp. (**C**) Overview of the real-time system.

**Table 1 sensors-21-04642-t001:** The performance of different types of classifiers according to the training and test dataset.

Classifier	10-Fold Cross Validation				Test Set Validation			
	Accuracy	Sensitivity	Specificity	AUC	Accuracy	Sensitivity	Specificity	AUC
LDA	0.88	0.83	0.92	0.93	0.80	0.74	0.87	0.90
Decision Tree	0.83	0.75	0.92	0.95	0.76	0.65	0.87	0.76
Linear SVM	0.90	0.88	0.92	0.92	0.80	0.74	0.87	0.90
RBF SVM	0.90	0.88	0.92	0.92	0.80	0.74	0.87	0.89

Note: LDA: linear discriminant analysis; Linear SVM: linear kernel support vector machine; RBF SVM: radial basis function kernel support vector machine.

**Table 2 sensors-21-04642-t002:** Performance comparison of the proposed method and previous methods of motion-sickness.

Study		Device	Feature	Classifier	Accuracy			
					Train Set	*n*	Test Set	*n*
1	Lin et al., 2013	HMD	EEG	PCA + SONFIN	0.82	17	-	-
2	Pane et al., 2018	LCD		CN2 Rules	0.89	9	-	-
3	Mawalid et al., 2018	LCD		Naïve Bayes	0.84	9	-	-
4	Li et al., 2020	HMD		SVM	0.79	18	-	-
5	Dennison Jr et al., 2019	HMD	EEG, EOG, RSP, etc.	Tree Bagger	0.95	18	-	-
6	Li et al., 2019	HMD	EEG and COP	Voting Classifier	0.76	20	-	-
7	Present study	HMD	Pupillary Response	SVM	0.90	48	0.80	46

Note: HMD: head-mounted display; LCD: crystal display liquid; EEG: electroencephalography; EOG: electrooculography; RSP: respiration; COP: center of pressure in force plate; PCA: principal component analysis; SONFIN: self-organizing neural fuzzy inference network; SVM: support vector machine.

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
