# Peer review of "Non-Contact Measurement of Motion Sickness Using Pupillary Rhythms from an Infrared Camera"

_sensors, 2021, doi:10.3390/s21144642_

Round 1

Reviewer 1 Report

The paper presents a procedure for measuring motion sickness, that appears as a side effect of viewing VR content, using pupillary rhythms and infrared images.

The work is very interesting, well organized, but some issues need to be fixed.

Main comments:

  • Introduction - Although infrared thermography does not represent the focus of the paper, it is a fundamental method in the presented procedure. Some information should be provided.
  • Materials and Methods – Provide the specification about the used infrared camera.
  • Figure 2 - Improve the resolution of the figure, provide the graph axes, replace second with s, and use more divisions in the axes scale.
  • Include an abbreviation section.
  • Report all unites (e.g. Lines 269, 270, 274, 275).
  • Figure 4 and 6 - Improve the resolution of the figure, replace second with s, and provide a legend of the specific signals.

Secondary comments

  • Improve the resolution of all figures.
  • Equation 1 – Provide a reference.
  • Line 146 – Replace “frames per second” with fps.
  • Line 147 – Provide the unit for 960 x 400.
  • Lines 159-161 – The infrared lamp has not been defined in the setup.
  • Line 210 – Do not report .05, but use the correct form (0.05). Check other cases (e.g. 256).
  • Figure 3, 5 and 7 – I do not understand the use of *** in the figures. Provide further explanation in the caption.
  • Appendix B should be Appendix A.

Reviewer 2 Report

Non-contact measurement of motion sickness using pupillary rhythms from an infrared camera

In this article, authors investigate non-contact measurement of motion sickness utilizing pupillary rhythms form an infrared camera.  Specifically, this research endeavor attempts to interpret the phenomenon of motion sickness and develop an advanced non-contact measurement method via an infrared webcam. Twenty-four subjects participated in a two-dimensional virtual reality and head-mounted devices. After a comprehensive analysis of collected data, authors directly stated that the proposed method was proven to capable of quantitatively measuring and monitoring motion sickness in real-time in a simple, economical, and contactless manner using an infrared camera.

The proposed unique advanced method of using pupillary rhythms via infrared camera is plausible. In details, the introduction (plus literature review) provides sufficient background with relevant references. However, the following reference needs to be included in this article.

Park, S., Lee, D. W., Mun, S., Kim, H. I., & Whang, M. (2018). Effect of Simulator Sickness Caused by Head-mounted Display on the Stability of the Pupillary Rhythm. Science of Emotion and Sensibility21(4), 43-54.

The research design and methodology of this investigation are appropriate and adequately described. Results are clearly presented with sufficient data analysis and plenty of relevant and clear tables and graphs (extensive figures). The conclusions are clearly presented and directly support the results. Overall, this article is a novel and original proposed method with significant content and scientifically sound implementation and presentation.

Reviewer 3 Report

The authors should put some examples of motion sickness in the abstract such as visual fatigue, anxiety, and nausea.

The authors should write the volunteer requirements in a toggle bulleted list for better visual acquisition.

The word score is wrong typed on equation (1).

An example of a questionnaire could be helpful in an appendix. To help the reader to understand how the volunteer assigns the symptoms. 

The authors must explain why each of the three factors that compose the SSQ has exactly seven items.

The appendix should start from letter A.

The phrase must start with a capital letter on line 163.

Why (x_0, y_0) instead of (x0,y0) represents the center position of the pupil on line 163.

Figure 2 could have another item order, item E after item C and item F after item D, because items E and F describe items C and D, respectively.

The Figure 2 description from lines 176 to 194 should have a relation between the step number to the item letter of Figure 2. 

The authors should plot a flowchart for the classification process explanation.  

In Figures 4 and 6, participant 6 has his or her identification ticked. In the same way, participants 1, 10, and 24 also would be identified.

Even though Bonferroni is a well-known mathematician, without a reference about Bonferroni's correction sounds that something seems to be missing.

From lines 344 to 349, there is no need to repeat the values presented in Table 1.

The authors must rewrite the first paragraph of Experimental Design because it is too similar to the Participants subsection of PARK and WHANG (2018). Other parts have similarities to the same article and need revision. 

PARK, Sangin; WHANG, Mincheol. Infrared camera-based non-contact measurement of brain activity from pupillary rhythms. Frontiers in physiology, v. 9, p. 1400, 2018.

Round 2

Reviewer 3 Report

The word score is wrong typed on equation (1).

The graphs C and D and E and F in Figure 2 should be horizontally aligned.